# Exploring perceptions of self-stigma of substance use and current alcohol and marijuana use patterns among college students

**Victoria O. Chentsova[1], Adrian J. Bravo[1]\*, Eleftherios Hetelekides[2], Daniel Gutierrez[3], Mark A. Prince[2], Stimulant Norms and Prevalence (SNAP) Study Team[¶]**

1 Department of Psychological Sciences, William & Mary, Williamsburg, Virginia, United States of America,
2 Department of Psychology, Colorado State University, Fort Collins, Colorado, United States of America,
3 Counseling and Special Education, School of Education, Virginia Commonwealth University, Richmond, Virginia, United States of America

¶ Membership of the Stimulant Norms and Prevalence (SNAP) Study Team is provided in the Acknowledgments.
* ajbravo@wm.edu

**Data Availability Statement:** Data and analytic outputs are available at https://doi.org/10.17605/OSF.IO/H8B59.

## Abstract

### Background

While research has examined the effect of stigma from others towards individuals with substance use disorders, few studies have examined the relationship between perceived self-stigma and engagement in substance use more broadly, especially among non-clinical samples.

### Aims

The present study examined the relationships between perceptions of self-stigma if one were to develop a substance use disorder, consisting of negative self-esteem and negative self-efficacy, and alcohol or marijuana use behaviors and outcomes.

### Method

Participants ($n = 2,243$; 71.5% female) were college students within the U.S. recruited to participate in an online survey on substance use with a special focus on alcohol and marijuana.

### Results

Results indicated no significant differences in stigma scores across individuals with different lifetime alcohol and marijuana use. Stigma ratings did differ between individuals with different profiles of last 30-day alcohol and marijuana use where, generally, individuals with lifetime use but no use in the last 30-day reported higher levels of self-stigma. Correlation analyses indicated that perceived impact of substance use disorder on sense of self-efficacy and self-esteem related negatively to nearly all observed factors of alcohol and marijuana use.

**Funding:** The author(s) thank the Office of the Provost of The College of William & Mary for a faculty summer research grant to Dr. Bravo in support of this work. The funders had no role in study design, data collection and analysis, decision to publish, or preparation of the manuscript.

**Competing interests:** The authors have declared that no competing interests exist.

## Conclusions

Though self-stigma, and stigma more broadly, has been shown to have negative implications for people with substance use disorders, the present study suggests that for non-clinical populations there may be some protective association between perceived self-stigma and alcohol or marijuana use engagement. This is not to say that self-stigma is a positive clinical intervention. Rather, we interpret these findings to indicate that negative perceptions of substance use disorder on the sense of self may be associated with distinct alcohol and marijuana use behaviors among young adults.

## Introduction

Within the literature on substance use stigma, negative stereotypes result predominately from public stigmatization [1, 2]. Public stigma leads to social segregation and exclusion and less favorable outcomes for individuals engaging in problematic substance use [3, 4]. Research has predominately examined the effect of perceived stigma from others towards someone with a substance use disorder (SUD) [5–9]. However, less research has examined the role of self-stigma associated with *developing* a SUD, let alone perceived self-stigma in individuals at a pre-SUD level of substance use. While various definitions of self-stigma exists, we use the definition by Kulesza and colleagues [10] that defines self-stigma as "as negative thoughts, feelings, and diminished self-image resulting from identification with the stigmatized group and anticipation of rejection from the larger society". Kulesza and colleagues review of the stigma-SUD literature concluded that "we have a very limited understanding of the relationship between substance use variables and stigma and are not able to conclusively say whether that relationship is linear in nature, positive, negative, or in what order of temporal precedence the relationship between substance use and stigma occurs" [10, p.10]. Although there is some research on self-stigma and substance misuse [11–13], little research has specifically examined perceived self-stigma for developing a SUD and substance use behaviors, especially among pre-clinical young adults.

Alcohol is among the top substances used by college-aged individuals (ages 18–25), with nearly 50.1% of people reporting using in the last month and nearly 15.0% of people meeting criteria for problematic use or AUD [14]. Marijuana use is also highly prevalent among young adults [15] and its excessive use has been shown to be associated with an array of short- and long-term consequences [16–18]. Although most young adults believe that marijuana is a low-risk substance, especially compared to other drugs [19, 20], some recognize that problematic marijuana use may culminate in a CUD [17, 21].

The influence of self-stigma, however, on alcohol and marijuana use outcomes in this population is not clear. Self-stigma is not uniform among individuals with substance use problems, let alone among individuals engaging in substance use without problems. The theory of planned behavior [22] posits that one's cognition of a behavior plays a role in their intention and use of that behavior. Accordingly, one may infer that higher reports of self-stigma for developing an AUD/CUD would lead to lower alcohol and marijuana use to avoid the negative effects of stigmatization (both self-directed and others-directed) if one was to develop a SUD. Specifically, it is possible that for individuals concerned with developing a SUD (i.e., those with negative cognition about using drugs problematically), this self-stigma may deter their motivations to use or may motivate controlled use altogether, which in turn would relate to lower use and consequences related to alcohol or marijuana.

## Purpose of the present study

In this study, we aimed to augment the understanding of college students' perceptions of perceived self-stigma associated with developing a SUD. Specifically, we aimed to: 1) examine mean differences in perceptions of self-stigma between individuals of different lifetime alcohol and marijuana use profiles (i.e. No Lifetime Use, Lifetime Alcohol only, Lifetime Marijuana Use only, Lifetime Alcohol and Marijuana Use); 2) among individuals endorsing some lifetime alcohol or marijuana use, examine mean differences in perception of self-stigma between individuals of different past 30-day alcohol and marijuana use profiles (i.e. No Use, Alcohol Only, Marijuana Only, Alcohol and Marijuana); and 3) examine if alcohol and marijuana use frequency, quantity, consequences, and use disorder symptoms, relate to self-perceptions of self-stigma. Using theory of planned behavior as a theoretical framework, we hypothesized that stigma scores would be significantly higher in individuals reporting no alcohol or marijuana use (both over the lifetime and within the past 30 days) compared to other use profiles. Further, we hypothesized that individuals with higher reported use frequency, quantity, experiences of negative consequences, and disorder symptoms, would report lower self-stigma perceptions of perceived SUD.

## Method

### Participants and procedure

Participants were college students recruited to participate in an online survey (standardized across sites) from students registered in Psychology Department Participant Pools at seven universities across six U.S. states (Colorado, New Mexico, New York, Virginia [two sites], Texas, and Wyoming) between fall 2019 and spring 2020. More information on the procedure for this study can be found in work by Looby and colleagues [23]. To minimize the burden on participants, we used a planned missing data design (i.e., matrix sampling), which has been used in other large multi-site college student studies [24]. Due to our missing-data-by-design procedure, the analytic sample for the present study was limited to 2,243 students who completed the measure of perceived self-stigma of SUD. Among our analytic sample, 47.6% ($n$ = 1,066) identified as White, non-Hispanic, the majority of students indicated that they were assigned female sex at birth (n = 1,599, 71.5%) and reported a mean age of 19.93 years ($SD$ = 3.57). Participants were provided a consent form to review within the online survey (i.e., first page of the survey was the consent form) and agreed to participate by clicking the "next" button (i.e., written consent). Participants received research participation credit for completing the study. This study used a single-site IRB and the University of Wyoming IRB approved all procedures.

### Measures

**Self-stigma of SUD.**   We assessed perceptions of self-stigma of SUD by adapting the Self-Stigma of Mental Illness Scale (SSOMI) created by Tucker and colleagues [25]. The SSOMI is used to address an individual's perception of how they would self-stigmatize if they were to identify with a particular identity. To account for our study intentions, we replaced the term "mental illness" with "substance use disorder" throughout the measure. Participants were presented with a set of ten statements regarding the degree to which they believed that their self-image would diminish if they were to have an SUD (e.g., "I would feel inadequate if I had a substance use disorder", "It would make me feel inferior to have a substance use disorder"). Participants were instructed to use a five-point scale ranging from 1 (*strongly disagree*) to 5 (*strongly agree*). An exploratory factor analysis (EFA) suggested that a two-factor model

solution fit the data best (see S1 Appendix for details of the EFA and items). Specifically, the measure consisted of two subscales: perceptions of negative self-esteem (NSE) concerned with how an individual would feel about themselves and perceptions of negative self-efficacy (NSEF) concerned with how an individual believes they would be able to control their behaviors. Items were summed and higher scores on each subscale indicate greater perceptions of NSE and NSEF.

**Alcohol and marijuana use behaviors.** Participants were asked to indicate if throughout their lifetime they ever consumed marijuana or alcohol. Among participants who endorsed lifetime alcohol and/or marijuana use, they were then asked how many days in the past month they consumed alcohol or marijuana. Among students who endorsed past 30-day marijuana use, typical marijuana use frequency/quantity were assessed using the Marijuana Use Grid [26]. Specifically, each day of the week was broken down into six 4-hour blocks of time (12a-4a, 4a-8a, 8a-12p, etc.), and participants were asked to report at which times they used marijuana during a "typical week" in the past 30 days, as well as the quantity of grams consumed during that time block. Among students who endorsed past 30-day alcohol use, we used a similar grid except participants were asked to report at which times they used alcohol during a "typical week" in the past 30 days as well as the quantity in standard drinks (e.g. 12 oz beer, 5 oz wine, 1.5oz shot of liquor) consumed during that time block. As done in previous work (23), we calculated typical frequency of alcohol/marijuana use by summing the total number of time blocks for which they reported using during the typical week (ranges: 0–42). We calculated typical quantity of alcohol/marijuana use by summing the total number of drinks/grams consumed across time blocks during the typical week (quantity estimates >3SDs above the mean were Winsorized).

**Alcohol and marijuana use outcomes.** Past 30-day marijuana-related problems were assessed using the 21-item Brief Marijuana Consequences Questionnaire (B-MACQ) [27] and the Cannabis Use Disorders Identification Test-Revised (CUDIT-R) [28]. The B-MACQ is a 21-item questionnaire that measures marijuana-related negative experiences within the past 30 days (e.g. "I have driven a car when I was high", "I have lost motivation to do things because of my marijuana use."). Participants were instructed to respond to statements regarding marijuana use consequences with "*Yes*" for the items they endorsed in the last 30-days and "*No*" for items they did not endorse in the last 30-days. A composite score reflective of the total number of distinct marijuana problems experienced in the past 30 days was created by summing all endorsed experiences. The CUDIT is a widely used measure based on the DSM diagnostic criteria for CUD, providing insight into an individual's endorsement of symptoms of CUD. Participants were presented with a series of 8 statements regarding their marijuana consumption behaviors (e.g. "How often do you use cannabis?", "How often during the past 6 months did you fail to do what was normally expected from you because of using cannabis?") and corresponding scales. A total score was calculated per the standard procedure recommended for the measure, by summing the scores across the items.

Past 30-day alcohol-related problems were assessed using the 24-item Brief-Young Adult Alcohol Consequences Questionnaire (B-YAACQ) [29] and the Alcohol Use Disorder Identification Test–US (AUDIT-US) [30]. The B-YAACQ is a 24-item questionnaire that measures alcohol-related negative experiences within the past 30 days (e.g., "I have often found it difficult to limit how much I drink", "When drinking, I have done impulsive things that I regretted later"). Like with the B-MACQ, participants were instructed to respond to statements regarding alcohol use consequences with "*Yes*" for the items they endorsed in the last 30-days and "*No*" for items they did not endorse in the last 30-days and then a composite score reflective of the total number of distinct alcohol problems experienced in the past 30 days was created by summing all endorsed experiences. The AUDIT, like the CUDIT, is a widely used measure

based on the DSM diagnostic criteria for AUD, providing insight into an individual's endorsement of symptoms of AUD. As with the CUDIT, participants were presented with a series of 10 statements regarding their drinking behaviors (e.g. "How often do you have a drink containing alcohol?", "Have you or someone else been injured because of your drinking?") and seven, five, or three answer choices depending on the scope of the question. A total score was calculated per the standard procedure recommended for the measure, by summing the scores across the items.

## Statistical analyses

Two ANOVAs were conducted using IBM SPSS Statistics (Version 27). The first ANOVA examined differences in self-stigma ratings between individuals with no lifetime alcohol or marijuana use, alcohol only, marijuana only, and alcohol and marijuana. The second ANOVA examined differences in self-stigma ratings between individuals with no past 30-day use, alcohol only in the last 30 days, marijuana only in the last 30 days, and alcohol and marijuana in the last 30 days (among lifetime users). Finally, correlations were examined to assess relationships between heightened self-stigma ratings and various problematic substance use outcomes. Given the large sample size and power of the present analyses, statistical significance was set at $p < .01$ for all analyses. For both ANOVAs, significant differences were further determined by a Bonferroni correction. Data and analytics outputs can be accessed at https://doi.org/10.17605/OSF.IO/H8B59.

## Results

### Perceptions of NSE and NESF across alcohol and marijuana use profiles

Prevalence rates of alcohol and marijuana use patterns and ANOVA results are presented in Table 1. In examining lifetime use ($n$ = 2,243), the ANOVA revealed no statistically significant difference between lifetime alcohol and marijuana use profiles on either self-stigma variable. In examining differences among past 30-day use profiles ($n$ = 2,009), we found significant differences for both NSEF ($F$[3, 2005] = 9.79, $p < .001$, $\eta 2$ = .014) and NSE $F$(3, 2005) = 6.03, $p < .001$, $\eta 2$ = .009) stigma variables. Individuals who had not used any alcohol or marijuana in the last 30 days reported significantly higher NSEF scores than any other group of individuals who used alcohol or marijuana. Further, individuals who had not used any alcohol or marijuana in

**Table 1. ANOVA results across substance use groups.**

| | Lifetime Substance Use ANOVA Results (n = 2, 243) | | | | |
|---|---|---|---|---|---|
| | None | Alcohol Only | Marijuana Only | Alcohol and Marijuana | Significant Differences |
| Sample Sizes | N = 234 | N = 749 | N = 22 | N = 1238 | - |
| Self- Efficacy | 17.27 (5.50) | 18.03 (4.94) | 17.32 (5.49) | 17.91 (4.82) | N/A |
| Self-Esteem | 19.95 (4.08) | 19.88 (3.98) | 18.00 (4.46) | 19.48 (4.21) | N/A |
| | 30-Day Substance Use ANOVA Results (n = 2, 009) | | | | |
| | None | Alcohol Only | Marijuana Only | Alcohol and Marijuana | Significant Differences |
| Sample Sizes | N = 436 | N = 894 | N = 94 | N = 585 | - |
| Self- Efficacy* | 19.06 (4.47) | 17.64 (4.97) | 17.50 (5.28) | 17.67 (4.82) | None > all other groups |
| Self-Esteem* | 19.96 (3.99) | 19.85 (4.00) | 18.90 (4.70) | 19.10 (4.30) | None > Alcohol and Marijuana<br>Alcohol Only > Alcohol and Marijuana |

*Note*. Given the large sample size and power of this analysis, we set alpha to < .01 for significance testing.

* Indicates $p < .01$. Significant differences in stigma measures across past substance use groups were determined by a Bonferroni corrected post-hoc comparisons.

**Table 2. Bivariate correlations among study variables.**

| | | | | | | | | |
|---|---|---|---|---|---|---|---|---|
| *Bivariate correlations among alcohol use variables* | | | | | | | | |
| | 1 | 2 | 3 | 4 | 5 | 6 | M | SD |
| 1. NSEF | <u>.84</u> | | | | | | 17.95 | 4.87 |
| 2. NSE | **.30** | <u>.88</u> | | | | | 19.61 | 4.14 |
| 3. Alcohol Use Frequency | -.03 | -.06 | - | | | | 3.55 | 3.07 |
| 4. Alcohol Use Quantity | **-.08** | **-.12** | **.76** | - | | | 9.74 | 9.46 |
| 5. BYAACQ | **-.10** | **-.11** | **.30** | **.44** | <u>.91</u> | | 5.41 | 5.31 |
| 6. AUDIT | **-.08** | **-.11** | **.44** | **.60** | **.64** | <u>.82</u> | 8.05 | 6.25 |
| *Bivariate correlations among marijuana study variables* | | | | | | | | |
| | 1 | 2 | 3 | 4 | 5 | 6 | M | SD |
| 1. NSEF | <u>.84</u> | | | | | | 17.95 | 4.87 |
| 2. NSE | **.30** | <u>.88</u> | | | | | 19.61 | 4.14 |
| 3. Marijuana Use Frequency | **-.15** | **-.19** | - | | | | 6.80 | 7.81 |
| 4. Marijuana Use Quantity | **-.11** | **-.15** | **.74** | - | | | 6.92 | 10.72 |
| 5. MACQ | -.06 | **-.14** | **.41** | **.30** | <u>.89</u> | | 4.31 | 4.52 |
| 6. CUDIT | **-.07** | **-.20** | **.59** | **.41** | **.66** | <u>.87</u> | 6.31 | 6.57 |

*Note*: Significant associations are in bold typeface for emphasis and were determined by $p < .01$. Cronbach's alpha for each measure is underlined and shown on the diagonal. NSE = perceptions of negative self-esteem; NSEF = perceptions of negative self-efficacy; BYAACQ = Brief Young Adult Consequences Questionnaire; AUDIT = Alcohol Use Disorder Identification Test; MACQ = Marijuana Consequences Questionnaire; CUDIT-R = Cannabis Use Disorder Identification Test.

the last 30 days or who had used alcohol only scored significantly higher on NSE than individuals who used both.

## Correlations with alcohol and marijuana and use outcomes

Bivariate correlations and descriptive statistics among study variables are reported in Table 2. At the bivariate level, NSE was significantly associated with NSEF ($r = .30$). Looking at alcohol specifically, NSEF was significantly negatively correlated with standard quantity consumed ($r = -.08$), B-YAACQ scores ($r = -.10$), and AUDIT scores ($r = -.08$). NSE was similarly negatively correlated with quantity of alcohol consumption ($r = -.12$), BYAACQ scores ($r = -.11$), and AUDIT scores ($r = -.11$). Taken together these findings suggests that higher perceptions of self-efficacy and self-esteem stigma were associated with lower quantity and problems associated with alcohol.

For marijuana outcomes, NSEF was shown to negatively correlate significantly with frequency ($r = -.15$), quantity ($r = -.11$), and CUDIT scores ($r = -.07$). NSE was negatively correlated with frequency ($r = -.19$), quantity ($r = -.15$), MACQ scores ($r = -.14$), and CUDIT scores ($r = -.20$). Taken together these findings suggests that higher perceptions of self-efficacy and self-esteem stigma were associated with lower frequency/quantity and problems associated with marijuana.

## Discussion

In this study, we examined the relationships of self-stigma perceptions of SUD with alcohol and marijuana use patterns. Though self-stigma of SUD did not differ between individuals of different lifetime alcohol and marijuana status, stigma ratings did differ between individuals with different profiles of last 30-day use (No Use, Alcohol Only, Marijuana Only, Alcohol and Marijuana) where, generally, individuals endorsing no alcohol or marijuana use in the last 30 days reported higher levels of self-stigma. Furthermore, we examined the correlations between

self-stigma and various alcohol and marijuana use practices. Findings indicated that perceived impact of SUD on sense of self-efficacy and self-esteem related negatively to nearly all observed factors of alcohol and marijuana use including frequency, quantity, negative consequences, and use disorder symptoms.

One possible explanation is that higher levels of SUD self-stigma in individuals who do not have SUD may activate an avoidance motive, or motive to educate and closely monitor oneself about substance use that is protective against SUDs [6]. A recent review explored the efficacy of "fear-based" messaging around SUD and found mixed results [31]. Other research has found that, on the other hand, favorable attitudes and expectancies relating to substance use can relate to problematic substance use down the line [32]. This is not to say that somehow self-stigma is a positive clinical intervention. In fact, previous research demonstrated that self-stigma and public stigma alike reduces the likelihood that those who are struggling with substance use will seek help [11]. Rather, we interpret these findings to indicate that increased awareness of the consequences of SUD on the sense of self may have an impact on *problematic* alcohol and marijuana use.

## Limitations & future research

These noteworthy implications should be taken with consideration of the study limitations. Whereas, we conducted our own psychometric exploration of the Self-Stigma scale used in this study, the scale is new and needs further refinement. For example, the measure was adapted to assess SUDs broadly and differing findings could occur if the measure was adapted to specific substances (e.g., alcohol use disorder or cannabis use disorder). We encourage future research on the validity and reliability of the Self-Stigma instrument. Further, the majority of our sample consisted of students' assigned female at birth. As a large body of existing research suggests that there are significant gender differences in stigmatization of substance use [33, 34], further analyses are needed to explore these potential differences in self-stigmatization.

It is also important to highlight that this research was intentionally focused on a non-clinical college population and thus cannot be extrapolated to broader populations. Though existing research point towards self-stigma being harmful in individuals with active substance use disorders [6] the concept of self-stigma is also an important area for continued research. There is need for future research exploring the unique relationships between stigma (including self-stigma) and substance use in non-clinical populations posited by this study. For example, understanding how self-concept and self-valuation occurs for emerging adults could lead to improvements in wellness programs, prevention initiatives, intervention for motivation, and overall new insights into addiction treatment. Further, the possibility of the presence of reporting bias should be of note for future research. Self-reports of stigma may have been complicated as the distinction between SUD and pre-clinical substance use was not explicitly defined to participants in this study. Similarly, the presence of SUD related stigma itself may have primed under- or over- reporting of substance use.

It is important that professionals focused on substance use are aware of the nuances in the relationship between stigma and SUD, particularly with a focus on problematic alcohol and marijuana use. On the one hand, existing literature has established that for individuals at the clinical level of problematic use–that is, a diagnosed AUD or CUD–stigma of the self as well as public stigma can be harmful in recovery efforts, especially in treatment seeking [35, 36], treatment retention [1, 37], and even relapse [38] as stigma can have major impact on self-esteem and self-efficacy. These areas of self-concept and personal devaluing could be important targets of therapeutic work. For individuals who have engaged in substance use and are now focusing

on sustaining recovery addressing self-stigma could be critical to relapse prevention. Thus, it is important for public health officials and clinical work done at the community level to be aware of stigmatizing messaging for these reasons.

The present study findings reveal that some level of self-stigma at the pre-clinical level–that is, pre-diagnosis of problematic alcohol or cannabis use–is associated with lower levels of substance use, substance use consequences, and substance use disorder symptoms. It is possible that there may be some *aspect* of self-stigma that is protective. Future studies can examine these relationships more deeply to establish causality, how self-stigma can influence substance use and related behavioral or functional outcomes longitudinally (e.g., education completion), and identify the *specific* factors within self-stigma that may be protective so that clinicians and public health experts alike can leverage those factors specifically in campaigns and legislature averting development of SUD. Future research may also try to explicitly identify distinct conditional associations of perceived self-stigma with substance use practices between clinical and non-clinical populations, as well as strata within non-clinical populations (e.g., people who use heavily daily, people who use lightly monthly).

It is critical to note here too that for clinicians and public health officials, understanding how to address the condition of non-clinical substance users and prevention efforts must be done carefully and with consideration to the residual effects that may have on individuals with substance use disorders, especially if targeting efforts on social and cultural level.

## Conclusions

Though self-stigma, and stigma more broadly, has been shown to have negative implications for people with substance use disorders, the present study suggests that for non-clinical populations there may be some protective association between self-stigma and substance use engagement. These findings establish grounds for further work in understanding what aspects of self-stigma may be protective or serve as predictive factors in engaging in problematic substance use. As college students and emerging adults make decisions to engage in substance use, it is important that they consider the consequences of their use, and the social and cultural contexts around them adapts to facilitate environments most conducive to unproblematic use. It is further important to understand the factors that may play into prevention of substance use disorders.

## Supporting information

**S1 Appendix. EFA results of the self-stigma of substance use disorder scale.**
(DOCX)

## Acknowledgments

This work could not have been completed without the efforts of the Stimulant Norms and Prevalence (SNAP) Study Team, which includes the following investigators (in alphabetical order): Adrian J. Bravo, William & Mary (Co-PI); Bradley T. Conner, Colorado State University; Mitch Earleywine, University at Albany, State University of New York; James Henson, Old Dominion University; Alison Looby, University of Wyoming (Co-PI); Mark A. Prince, Colorado State University; Ty Schepis, Texas State University; Margo Villarosa-Hurlocker, University of New Mexico.

## Author Contributions

**Conceptualization:** Victoria O. Chentsova, Adrian J. Bravo, Mark A. Prince.

**Data curation:** Adrian J. Bravo, Mark A. Prince.

**Formal analysis:** Adrian J. Bravo, Mark A. Prince.

**Funding acquisition:** Adrian J. Bravo.

**Investigation:** Adrian J. Bravo.

**Methodology:** Adrian J. Bravo, Mark A. Prince.

**Project administration:** Adrian J. Bravo.

**Resources:** Adrian J. Bravo, Mark A. Prince.

**Supervision:** Adrian J. Bravo.

**Writing – original draft:** Victoria O. Chentsova, Adrian J. Bravo, Eleftherios Hetelekides, Daniel Gutierrez, Mark A. Prince.

**Writing – review & editing:** Victoria O. Chentsova, Adrian J. Bravo, Eleftherios Hetelekides, Daniel Gutierrez, Mark A. Prince.

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
