## [Decision Letter · Decision Letter 0]

19 Apr 2023

PONE-D-23-05413Exploring Perceptions of Self-Stigma of Substance Use and Current Substance Use Patterns among College StudentsPLOS ONE

Dear Dr. Bravo,

Thank you for submitting your manuscript to PLOS ONE. After careful consideration, we feel that it has merit but does not fully meet PLOS ONE’s publication criteria as it currently stands. Therefore, we invite you to submit a revised version of the manuscript that addresses the points raised during the review process.

We look forward to receiving your revised manuscript.

Kind regards,

Boshra Ismael Ahmed Arnout

Academic Editor

PLOS ONE

Journal Requirements:

Additional Editor Comments:

Dear Author

The paper PONE-D-23-05413 has been reviewed by experts in the field who consider that the paper can publish after major revision.

We wish you a meaningful day.

Yours Sincerely

Reviewers' comments:

Reviewer's Responses to Questions

**Comments to the Author**

1. Is the manuscript technically sound, and do the data support the conclusions?

Reviewer #1: Yes

2. Has the statistical analysis been performed appropriately and rigorously? 

Reviewer #1: Yes

3. Have the authors made all data underlying the findings in their manuscript fully available?

Reviewer #1: No

4. Is the manuscript presented in an intelligible fashion and written in standard English?

Reviewer #1: Yes

5. Review Comments to the Author

Reviewer #1: This study examined the relationship between perceived self-stigma related to substance use disorders and engagement in substance use among non-clinical populations. The results showed that individuals with lifetime use but no substance use in the last 30-day reported higher levels of self-stigma. Perceived impact of SUD on sense of self-efficacy and self-esteem related negatively to nearly all observed factors of alcohol and marijuana use. The study suggests that increased awareness of the consequences of SUD on the sense of self has an impact on alcohol and marijuana use among young adults.

Overall, the background is well-written and provide a clear overview of the current state of knowledge regarding substance use stigma and the specific research questions addressed in the present study. The background also effectively highlights the gap in the literature regarding the role of self-stigma in the development of SUDs and the need for more research in this area. The background also provides a good introduction to the study's hypotheses and research questions, outlining the specific aims of the study and how they relate to the broader literature on substance use stigma. The use of the theory of planned behavior as a framework for understanding the relationship between self-stigma and substance use outcomes is a useful approach and shows that the study is grounded in a strong theoretical foundation. The background provides appropriate citations to support the claims made, and the use of direct quotes from previous literature adds credibility to the discussion.

The authors did good job in outlining their methods, result, and discussion. However, based on the information provided, here are some recommendations for the authors:

- The authors discuss their findings within the scope of SUD; however, they only measured alcohol and marijuana. It is not scientifically sound to pinpoint two substances only and generalize it to SUD. My recommendation is to revise your title and hypothesis to indicate in within alcohol and marijuana instead of SUD.

- Clarify the study design and sampling method: The authors should provide more information on how participants were recruited, the inclusion/exclusion criteria, and any potential biases in the sample.

- Provide more detail on the measures used: While the authors briefly described the measures used, it still misses important information such as the operational definitions and elements of NSE, NSEF, B-MACQ; CUDIT-R, etc.

- Discuss implications and future directions: While the authors briefly discussed the implications of their findings, they could provide a more in-depth discussion of how their results could inform future research or interventions for substance use disorders. Additionally, they could discuss any potential limitations of their study and areas for future research.

Thank you for doing such important work.

6. PLOS authors have the option to publish the peer review history of their article (what does this mean?). If published, this will include your full peer review and any attached files.

Reviewer #1: No

---

## [Author Response · Author response to Decision Letter 0]

13 Jun 2023

Dr. Boshra Ismael Ahmed Arnout,

We would like to thank you for the quick and thorough review of our manuscript and for the selection of well-qualified reviewers. We have addressed the reviewer’s concerns and summarized our changes below. By addressing the general journal requirements and reviewers’ comments, we believe our manuscript has been strengthened and will make more of a contribution to the field.

Journal (s)' Comments: 

** The manuscript was revised to fit PLOS ONE style requirements per the links provided above and saved as “Revised Manuscript with Track Changes” and “Manuscript” for the track changes and clean versions of the document, respectively. 

2. Please provide additional details regarding participant consent. In the ethics statement in the Methods and online submission information, please ensure that you have specified (1) whether consent was informed and (2) what type you obtained (for instance, written or verbal, and if verbal, how it was documented and witnessed). If your study included minors, state whether you obtained consent from parents or guardians. If the need for consent was waived by the ethics committee, please include this information. If you are reporting a retrospective study of medical records or archived samples, please ensure that you have discussed whether all data were fully anonymized before you accessed them and/or whether the IRB or ethics committee waived the requirement for informed consent. If patients provided informed written consent to have data from their medical records used in research, please include this information.

** The manuscript was revised to highlight additional detail on the informed consent systems employed for this study. 

** The funding information has been reviewed for consistency and entered in the portal accordingly.

4. In your Data Availability statement, you have not specified where the minimal data set underlying the results described in your manuscript can be found. 

PLOS defines a study's minimal data set as the underlying data used to reach the conclusions drawn in the manuscript and any additional data required to replicate the reported study findings in their entirety. All PLOS journals require that the minimal data set be made fully available. For more information about our data policy, please see http://journals.plos.org/plosone/s/data-availability. Upon re-submitting your revised manuscript, please upload your study’s minimal underlying data set as either Supporting Information files or to a stable, public repository and include the relevant URLs, DOIs, or accession numbers within your revised cover letter. For a list of acceptable repositories, please see http://journals.plos.org/plosone/s/data-availability#loc-recommended-repositories. Any potentially identifying patient information must be fully anonymized.

** The data availability statement has been added to the cover letter.

5. Please include your full ethics statement in the ‘Methods’ section of your manuscript file. 

In your statement, please include the full name of the IRB or ethics committee who approved or waived your study, as well as whether or not you obtained informed written or verbal consent. If consent was waived for your study, please include this information in your statement as well. 

** This information is provided in the Method section of the manuscript. 

** The supporting information has been revised within the manuscript to reflect journal requirements. 

Reviewer(s)' Response to Questions:

1. Is the manuscript technically sound, and do the data support the conclusions? 

Reviewer #1: Yes

** No action is required. 

2. Has the statistical analysis been performed appropriately and rigorously?

Reviewer #1: Yes

** No action is required. 

3. Have the authors made all data underlying the findings in their manuscript fully available?

Reviewer #1: No

** The URL to the data is provided in the Data Availability section of the submission portal.

4. Is the manuscript presented in an intelligible fashion and written in standard English?

Reviewer #1: Yes

** No action is required. 

Reviewer #1: This study examined the relationship between perceived self-stigma related to substance use disorders and engagement in substance use among non-clinical populations. The results showed that individuals with lifetime use but no substance use in the last 30-day reported higher levels of self-stigma. Perceived impact of SUD on sense of self-efficacy and self-esteem related negatively to nearly all observed factors of alcohol and marijuana use. The study suggests that increased awareness of the consequences of SUD on the sense of self has an impact on alcohol and marijuana use among young adults.

Overall, the background is well-written and provide a clear overview of the current state of knowledge regarding substance use stigma and the specific research questions addressed in the present study. The background also effectively highlights the gap in the literature regarding the role of self-stigma in the development of SUDs and the need for more research in this area. The background also provides a good introduction to the study's hypotheses and research questions, outlining the specific aims of the study and how they relate to the broader literature on substance use stigma. The use of the theory of planned behavior as a framework for understanding the relationship between self-stigma and substance use outcomes is a useful approach and shows that the study is grounded in a strong theoretical foundation. The background provides appropriate citations to support the claims made, and the use of direct quotes from previous literature adds credibility to the discussion. The authors did good job in outlining their methods, result, and discussion. However, based on the information provided, here are some recommendations for the authors:

** We appreciate the laudatory remarks. We thank the reviewer for their insightful comments, which we believe, improved our article significantly.

1) The authors discuss their findings within the scope of SUD; however, they only measured alcohol and marijuana. It is not scientifically sound to pinpoint two substances only and generalize it to SUD. My recommendation is to revise your title and hypothesis to indicate in within alcohol and marijuana instead of SUD.

** Thank you for this observation. We have revised the title and various portions of the manuscript to specify alcohol and marijuana rather than substances broadly. 

2) Clarify the study design and sampling method: The authors should provide more information on how participants were recruited, the inclusion/exclusion criteria, and any potential biases in the sample.

** As this manuscript is meant to be read as a brief report, details on design and sampling methods are kept general. However, we added some general information about the sample and also included the citation for another manuscript (and the URL per journal requirements) that provides greater detail on study design. 

3) Provide more detail on the measures used: While the authors briefly described the measures used, it still misses important information such as the operational definitions and elements of NSE, NSEF, B-MACQ; CUDIT-R, etc.

** As this manuscript is meant to be read as a brief report, details on design and sampling methods are kept general. Further information on the measures used in this study are also available in the manuscript referenced within the method section for additional information. However, we also added some additional detail about each of the relevant measures to the present research. 

4) Discuss implications and future directions: While the authors briefly discussed the implications of their findings, they could provide a more in-depth discussion of how their results could inform future research or interventions for substance use disorders. Additionally, they could discuss any potential limitations of their study and areas for future research.

** Thank you for these suggestions. Although this manuscript was intended to be read as a brief report, we have added additional text regarding implications, limitations, and future research.

---

## [Decision Letter · Decision Letter 1]

13 Nov 2023

PONE-D-23-05413R1Exploring perceptions of self-stigma of substance use and current alcohol and marijuana use patterns among college studentsPLOS ONE

Dear Dr. Bravo,

Thank you for submitting your manuscript to PLOS ONE. After careful consideration, we feel that it has merit but does not fully meet PLOS ONE’s publication criteria as it currently stands. Therefore, we invite you to submit a revised version of the manuscript that addresses the points raised during the review process.

We look forward to receiving your revised manuscript.

Kind regards,

Lakshit Jain, MD

Academic Editor

PLOS ONE

Journal Requirements:

**Additional Editor Comments:**

Thank you for revising your article and making edits as recommended.

Unfortunately, there are still some minor concerns that need to be addressed. Please specifically address concerns raised by reviewer 3 and 7.

Thanks 

Lakshit  

Reviewers' comments:

Reviewer's Responses to Questions

**Comments to the Author**

1. If the authors have adequately addressed your comments raised in a previous round of review and you feel that this manuscript is now acceptable for publication, you may indicate that here to bypass the “Comments to the Author” section, enter your conflict of interest statement in the “Confidential to Editor” section, and submit your "Accept" recommendation.

Reviewer #2: (No Response)

Reviewer #3: All comments have been addressed

Reviewer #4: All comments have been addressed

Reviewer #5: All comments have been addressed

Reviewer #6: All comments have been addressed

Reviewer #7: (No Response)

2. Is the manuscript technically sound, and do the data support the conclusions?

Reviewer #2: Yes

Reviewer #3: Yes

Reviewer #4: Yes

Reviewer #5: Yes

Reviewer #6: Yes

Reviewer #7: Yes

3. Has the statistical analysis been performed appropriately and rigorously? 

Reviewer #2: I Don't Know

Reviewer #3: Yes

Reviewer #4: Yes

Reviewer #5: Yes

Reviewer #6: I Don't Know

Reviewer #7: Yes

4. Have the authors made all data underlying the findings in their manuscript fully available?

Reviewer #2: Yes

Reviewer #3: Yes

Reviewer #4: Yes

Reviewer #5: Yes

Reviewer #6: Yes

Reviewer #7: Yes

5. Is the manuscript presented in an intelligible fashion and written in standard English?

Reviewer #2: Yes

Reviewer #3: Yes

Reviewer #4: Yes

Reviewer #5: Yes

Reviewer #6: Yes

Reviewer #7: Yes

6. Review Comments to the Author

Reviewer #2: This is a well-written manuscript. I liked reading this manuscript and believe that it is very promising. At the same time, I identified couple of issues that require the authors’ attention. It is interesting to see the correlation between pre-clinical substance use, stigma and its impact on individuals' future. Professional use of English language is at par. The manuscript is based on impressive empirical evidence and makes an original contribution but there should be some comment on possible bias like reporting bias of the study participants and also their understanding of the difference between pre-clinical substance uses and SUD itself. It will be very interesting to see the follow up study on these subject individuals regarding how many of these develops SUD.

Reviewer #3: The authors have addressed most of the review comments except as stated below. Once the pending comment is addressed the article may be accepted for publication.

1) The authors discuss their findings within the scope of SUD; however, they only

measured alcohol and marijuana. It is not scientifically sound to pinpoint two

substances only and generalize it to SUD. My recommendation is to revise your title

and hypothesis to indicate in within alcohol and marijuana instead of SUD.

The title is more clear and the hypothesis is now more sound

** 2) Clarify the study design and sampling method: The authors should provide more

information on how participants were recruited, the inclusion/exclusion criteria, and any

potential biases in the sample.

The authors have clarified this comment

3) Provide more detail on the measures used: While the authors briefly described the

measures used, it still misses important information such as the operational definitions

and elements of NSE, NSEF, B-MACQ; CUDIT-R, etc.

The authors have clarified this comment

4) Discuss implications and future directions: While the authors briefly discussed the

implications of their findings, they could provide a more in-depth discussion of how

their results could inform future research or interventions for substance use disorders.

Additionally, they could discuss any potential limitations of their study and areas for

future research.

The authors have clarified this comment. However, the current manuscript brings an important question. The authors themselves report how in pre problematic use population self stigma may prevent problematic use but in patients with Use Disorders they may serve as a deterrent to seek treatment. The authors may identify this as a study area where populations can be compared, or stratify their data and present the results of stigma in groups stratified on the basis of amount/frequency of use or substance of use (alcohol or cannabis).

This would make this paper more methodologically rigorous and increase the scientific impact of this paper.

I look forward to reading the revised manuscript.

Reviewer #4: Thank you for the opportunity to review this brief report that examines the relationship between self stigma related to alcohol and MJ use and engagement in use in college students.

The manuscript is well written, and succinct and the background section explains the rationale for the study well.

The findings are clearly communicated.

The authors seem to have addressed the concerns of the previous reviewer well making significant changes especially to the discussion section which was well written, especially the limitations.

The main finding of the study that shows that self stigma could potentially play a protective role against use is valuable and an important contribution to the literature.

I don’t have any other major edits/suggestions.

Great work.

Reviewer #5: The research paper details the relationship between the self-stigma of marijuana and alcohol use and engagement in these substances among college students. It Indicates that individuals with lifetime use of marijuana and alcohol, but no recent use reported higher self-stigma. Negative self-esteem and self-efficacy were found to relate negatively to nearly all observed factors of alcohol and marijuana use.

Feedback:

The assertion positing a link between self-stigma and involvement in substance use is substantiated by robust research findings, which reveal that individuals who have a history of substance use but no recent use exhibits elevated levels of self-stigma. This solid empirical basis lends credence to the argument, rendering it valid. Likewise, the claim proposing a negative relationship between negative self-esteem and self-efficacy with alcohol and marijuana use is bolstered by the study's outcomes. The argument finds strong empirical support in the research data, further establishing its validity. Moreover, the argument asserting that an enhanced awareness of the impact of substance use disorders on an individual's sense of self can influence alcohol and marijuana use is well-founded in the study's results. This logical reasoning harmonizes effectively with the study's objectives, thus underscoring its validity.

The authors have diligently responded to the concerns raised in the previous review and have effectively rectified the identified issues. Their commitment to addressing these concerns is evident in the revised manuscript. They have demonstrated a high level of responsiveness and professionalism in dealing with the feedback, which reflects positively on the overall quality of the research. The corrections made by the authors have notably improved the clarity and coherence of the manuscript. The revised version now presents a more refined and accurate portrayal of their research findings, strengthening the overall credibility and impact of their work. The authors' attention to detail and their willingness to make the necessary revisions exemplify their commitment to producing a high-quality academic contribution. This constructive engagement with feedback not only benefits the authors by enhancing the quality of their work but also contributes to the scholarly community by ensuring that accurate and reliable research is disseminated. It is evident that the authors' dedication to addressing the concerns raised in the previous review will enhance the value and relevance of their research within the academic field.

Reviewer #6: (No Response)

Reviewer #7: Thanks for giving me the opportunity to review this article. The article is interesting as it explores the protective effect of self stigma on alcohol and marijuana use. This is a rare concept that has not been explored much so far. The article is very unique in that aspect. The sample size is excellent. Also the author has been very candid in acknowledging the limitation of the study that majority of participants were females. It would be interesting to see the results of the study if all genders were included( Including LGBT etc)

2. One limitation of the study is the study included college students. There are a lot of students that drop out of high school due to substance use problems and it would be great to know how their self stigma compares to someone who has made it to college.

SOLUTION: Please include in the limitation that perhaps including the college going students have skewed the results of the study and the results may not be extrapolated to the entire population.

7. PLOS authors have the option to publish the peer review history of their article (what does this mean?). If published, this will include your full peer review and any attached files.

Reviewer #2: No

Reviewer #3: No

Reviewer #4: No

Reviewer #5: **Yes: **Aditi Sharma

Reviewer #6: No

Reviewer #7: **Yes: **Jasleen Kaur MD

---

## [Author Response · Author response to Decision Letter 1]

30 Dec 2023

Dr. Lakshit Jain,

We would like to thank you for the thorough review of our manuscript and for the selection of well-qualified reviewers. We have addressed the reviewer’s concerns and summarized our changes below. By addressing the general journal requirements and reviewers’ comments, we believe our manuscript has been strengthened and will make more of a contribution to the field.

Journal (s)' Comments/Requirements: 

** We have reviewed our reference list and assure that is complete and correct. Minor changes were made to the formatting of the references (e.g., writing references in sentence-case). Further, no studies cited have been redacted to the best of our knowledge.

Additional Editor Comments:

1. Thank you for revising your article and making edits as recommended. Unfortunately, there are still some minor concerns that need to be addressed. Please specifically address concerns raised by reviewer 3 and 7.

Thanks, Lakshit 

** We have addressed each reviewers’ concerns and summarized our changes below. By addressing the reviewer’s comments, we believe our manuscript has been strengthened and will make more of a contribution to the field.

Reviewer(s)' Response to Questions:

1. If the authors have adequately addressed your comments raised in a previous round of review and you feel that this manuscript is now acceptable for publication, you may indicate that here to bypass the “Comments to the Author” section, enter your conflict of interest statement in the “Confidential to Editor” section, and submit your "Accept" recommendation.

Reviewer #2: (No Response)

Reviewer #3: All comments have been addressed

Reviewer #4: All comments have been addressed

Reviewer #5: All comments have been addressed

Reviewer #6: All comments have been addressed

Reviewer #7: (No Response)

** No action is required. 

2. Is the manuscript technically sound, and do the data support the conclusions? 

Reviewer #2: Yes

Reviewer #3: Yes

Reviewer #4: Yes

Reviewer #5: Yes

Reviewer #6: Yes

Reviewer #7: Yes

** No action is required. 

3. Has the statistical analysis been performed appropriately and rigorously?

Reviewer #2: I Don't Know

Reviewer #3: Yes

Reviewer #4: Yes

Reviewer #5: Yes

Reviewer #6: I Don't Know

Reviewer #7: Yes

** No action is required. 

4. Have the authors made all data underlying the findings in their manuscript fully available?

Reviewer #2: Yes

Reviewer #3: Yes

Reviewer #4: Yes

Reviewer #5: Yes

Reviewer #6: Yes

Reviewer #7: Yes

** No action is required.

5. Is the manuscript presented in an intelligible fashion and written in standard English?

Reviewer #2: Yes

Reviewer #3: Yes

Reviewer #4: Yes

Reviewer #5: Yes

Reviewer #6: Yes

Reviewer #7: Yes

** No action is required. 

Review Comments to the Author:

Reviewer #2: This is a well-written manuscript. I liked reading this manuscript and believe that it is very promising. At the same time, I identified couple of issues that require the authors’ attention. It is interesting to see the correlation between pre-clinical substance use, stigma and its impact on individuals' future. Professional use of English language is at par. The manuscript is based on impressive empirical evidence and makes an original contribution but there should be some comment on possible bias like reporting bias of the study participants and also their understanding of the difference between pre-clinical substance uses and SUD itself. It will be very interesting to see the follow up study on these subject individuals regarding how many of these develops SUD.

** We appreciate the laudatory remarks. We thank the reviewer for their insightful comments, which we believe improved our article significantly. Regarding the issues about bias in reporting, we have now added this as a limitation within the “limitations and future research” section. Specifically, we now state, “Further, the possibility of the presence of reporting bias should be of note for future research. Self-reports of stigma may have been complicated as the distinction between SUD and pre-clinical substance use was not explicitly defined to participants in this study. Similarly, the presence of SUD related stigma itself may have primed under- or over- reporting of substance use” (see p. 13).

Reviewer #3: The authors have addressed most of the review comments except as stated below. Once the pending comment is addressed the article may be accepted for publication.

** We appreciate the laudatory remarks. We thank the reviewer for their insightful comments, which we believe improved our article significantly.

1) The authors discuss their findings within the scope of SUD; however, they only measured alcohol and marijuana. It is not scientifically sound to pinpoint two substances only and generalize it to SUD. My recommendation is to revise your title and hypothesis to indicate in within alcohol and marijuana instead of SUD.

The title is more clear and the hypothesis is now more sound

** We appreciate the feedback.

2) Clarify the study design and sampling method: The authors should provide more

information on how participants were recruited, the inclusion/exclusion criteria, and any

potential biases in the sample.

The authors have clarified this comment.

** We appreciate the feedback.

3) Provide more detail on the measures used: While the authors briefly described the

measures used, it still misses important information such as the operational definitions

and elements of NSE, NSEF, B-MACQ; CUDIT-R, etc.

The authors have clarified this comment.

** We appreciate the feedback.

4) Discuss implications and future directions: While the authors briefly discussed the

implications of their findings, they could provide a more in-depth discussion of how

their results could inform future research or interventions for substance use disorders.

Additionally, they could discuss any potential limitations of their study and areas for

future research.

The authors have clarified this comment. However, the current manuscript brings an important question. The authors themselves report how in pre problematic use population self stigma may prevent problematic use but in patients with Use Disorders they may serve as a deterrent to seek treatment. The authors may identify this as a study area where populations can be compared, or stratify their data and present the results of stigma in groups stratified on the basis of amount/frequency of use or substance of use (alcohol or cannabis).

This would make this paper more methodologically rigorous and increase the scientific impact of this paper. I look forward to reading the revised manuscript.

** The reviewer makes an astute point. Comparing perceived self-stigma scores across amount/frequency or substance of use could add to this paper; however, this would be complicated by the fact that our perceived self-stigma variable is based on “substance use disorder” broadly and making such comparisons may not be as methodologically sound. Nonetheless, this is an important point, and we are actively collecting data on a follow-up study where we actually break down “perceived self-stigma” to focus on alcohol and cannabis specific use disorders. Thus, within those new data we can better stratify and determine whether frequency/amount of use of that drug actually impacts perceived stigma of that specific drug use disorder. 

Regarding the comparisons between pre-clinical and clinical samples, we have further emphasized in the future directions section that more work comparing across pre-clinical and clinical samples is needed to determine the overall impact self-stigma (and stigma more broadly) has on substance use patterns. Specifically, we now state, “Future research may also try to explicitly identify distinct conditional associations of perceived self-stigma with substance use practices between clinical and non-clinical populations, as well as strata within non-clinical populations (e.g., people who use heavily daily, people who use lightly monthly)” (see p. 14).

Reviewer #4: Thank you for the opportunity to review this brief report that examines the relationship between self stigma related to alcohol and MJ use and engagement in use in college students. The manuscript is well written, and succinct and the background section explains the rationale for the study well. The findings are clearly communicated. The authors seem to have addressed the concerns of the previous reviewer well making significant changes especially to the discussion section which was well written, especially the limitations. The main finding of the study that shows that self stigma could potentially play a protective role against use is valuable and an important contribution to the literature. I don’t have any other major edits/suggestions. Great work.

** We appreciate the laudatory remarks. We thank the reviewer for their insightful comments, which we believe improved our article significantly.

Reviewer #5: The research paper details the relationship between the self-stigma of marijuana and alcohol use and engagement in these substances among college students. It Indicates that individuals with lifetime use of marijuana and alcohol, but no recent use reported higher self-stigma. Negative self-esteem and self-efficacy were found to relate negatively to nearly all observed factors of alcohol and marijuana use.

Feedback:

The assertion positing a link between self-stigma and involvement in substance use is substantiated by robust research findings, which reveal that individuals who have a history of substance use but no recent use exhibits elevated levels of self-stigma. This solid empirical basis lends credence to the argument, rendering it valid. Likewise, the claim proposing a negative relationship between negative self-esteem and self-efficacy with alcohol and marijuana use is bolstered by the study's outcomes. The argument finds strong empirical support in the research data, further establishing its validity. Moreover, the argument asserting that an enhanced awareness of the impact of substance use disorders on an individual's sense of self can influence alcohol and marijuana use is well-founded in the study's results. This logical reasoning harmonizes effectively with the study's objectives, thus underscoring its validity.

The authors have diligently responded to the concerns raised in the previous review and have effectively rectified the identified issues. Their commitment to addressing these concerns is evident in the revised manuscript. They have demonstrated a high level of responsiveness and professionalism in dealing with the feedback, which reflects positively on the overall quality of the research. The corrections made by the authors have notably improved the clarity and coherence of the manuscript. The revised version now presents a more refined and accurate portrayal of their research findings, strengthening the overall credibility and impact of their work. The authors' attention to detail and their willingness to make the necessary revisions exemplify their commitment to producing a high-quality academic contribution. This constructive engagement with feedback not only benefits the authors by enhancing the quality of their work but also contributes to the scholarly community by ensuring that accurate and reliable research is disseminated. It is evident that the authors' dedication to addressing the concerns raised in the previous review will enhance the value and relevance of their research within the academic field.

** We appreciate the laudatory remarks. We thank the reviewer for their insightful comments, which we believe improved our article significantly.

Reviewer #6: (No Response)

** We thank the reviewer for their review. 

Reviewer #7: Thanks for giving me the opportunity to review this article. The article is interesting as it explores the protective effect of self stigma on alcohol and marijuana use. This is a rare concept that has not been explored much so far. The article is very unique in that aspect. The sample size is excellent. Also the author has been very candid in acknowledging the limitation of the study that majority of participants were females. It would be interesting to see the results of the study if all genders were included( Including LGBT etc)

** We thank the reviewer for their insightful comments, which we believe improved our article significantly. We also agree that examining these relationships among diverse populations is a needed area of research in our future work and are committed to supporting expansion of research into this area of intersectional identities.

1. One limitation of the study is the study included college students. There are a lot of students that drop out of high school due to substance use problems and it would be great to know how their self stigma compares to someone who has made it to college.

SOLUTION: Please include in the limitation that perhaps including the college going students have skewed the results of the study and the results may not be extrapolated to the entire population.

** The reviewer makes an astute point. Regarding the issues about high school students and that our sample does not represent all college students, we have now added this as a limitation within the “limitations and future research” section. Specifically, we now state, “It is also important to highlight that this research was intentionally focused on a non-clinical college population and thus cannot be extrapolated to broader populations” (see p. 13). We also state, “Future studies can examine these relationships more deeply to establish causality, how self-stigma can influence substance use and related behavioral or functional outcomes longitudinally (e.g., education completion), and identify the specific factors within self-stigma that may be protective so that clinicians and public health experts alike can leverage those factors specifically in campaigns and legislature averting development of SUD” (see p.14).

---

## [Decision Letter · Decision Letter 2]

11 Jan 2024

PONE-D-23-05413R2Exploring perceptions of self-stigma of substance use and current alcohol and marijuana use patterns among college studentsPLOS ONE

Dear Dr. Bravo,

Thank you for submitting your manuscript to PLOS ONE. After careful consideration, we feel that it has merit but does not fully meet PLOS ONE’s publication criteria as it currently stands. Therefore, we invite you to submit a revised version of the manuscript that addresses the points raised during the review process.

We look forward to receiving your revised manuscript.

Kind regards,

Lakshit Jain, MD

Academic Editor

PLOS ONE

Additional Editor Comments:

Three reviewers have recommend to accept this article, but reviewer 8 is seeking this article for the first time, they have raised multiple concerns that should be resolved prior to accepting this article

Reviewers' comments:

Reviewer's Responses to Questions

**Comments to the Author**

1. If the authors have adequately addressed your comments raised in a previous round of review and you feel that this manuscript is now acceptable for publication, you may indicate that here to bypass the “Comments to the Author” section, enter your conflict of interest statement in the “Confidential to Editor” section, and submit your "Accept" recommendation.

Reviewer #2: All comments have been addressed

Reviewer #3: All comments have been addressed

Reviewer #5: All comments have been addressed

Reviewer #8: (No Response)

2. Is the manuscript technically sound, and do the data support the conclusions?

Reviewer #2: Yes

Reviewer #3: Yes

Reviewer #5: Yes

Reviewer #8: Partly

3. Has the statistical analysis been performed appropriately and rigorously? 

Reviewer #2: Yes

Reviewer #3: Yes

Reviewer #5: Yes

Reviewer #8: I Don't Know

4. Have the authors made all data underlying the findings in their manuscript fully available?

Reviewer #2: Yes

Reviewer #3: Yes

Reviewer #5: Yes

Reviewer #8: Yes

5. Is the manuscript presented in an intelligible fashion and written in standard English?

Reviewer #2: Yes

Reviewer #3: Yes

Reviewer #5: Yes

Reviewer #8: Yes

6. Review Comments to the Author

Reviewer #2: (No Response)

Reviewer #3: The authors have sufficiently and succinctly made appropriate changes to the manuscript to make it more reader friendly. The paper presents an interesting approach to alcohol and cannabis use disorder propagation and at this stage would benefit from acceptance in the journal. Thank you for the opportunity to review this paper.

Reviewer #5: The paper has improved after the revisions. The authors' meticulous efforts have substantially strengthened the manuscript, aligning it more closely with the standards of PLOS ONE

Reviewer #8: This manuscript presents information on college students’ perception of self- stigma associated with developing a substance use disorder (SUD) in the context of alcohol and marijuana use. It is well written and below are my suggestions on how the manuscript can be improved.

Abstract

Generally, the abstract would benefit from a structure; introduction of different sections will improve focus and clarity

Line 32

There is no mention of the instruments used to collect the data.

Line 34 – 40

The results in this section require the support of the numbers that tell the story. Presentation could benefit by stating the actual figures obtained in the statistical analysis, including the p values while mentioning the significance levels.

For instance, were the higher levels of self- stigma mentioned in Line 37- 38 statistically significant? What were the actual figures?

Line 44- 46

“Rather, we interpret these findings to indicate that increased awareness of the consequences of substance use disorder on the sense of self has an impact on alcohol and marijuana use among young adults”. When one has the objectives of the study in mind, it appears the different patterns of alcohol and marijuana use are actually the independent variables with self- stigma, and not awareness, as the dependent variable. This may require clarification.

The conclusion of “increased awareness” cannot emanate from the findings of this particular study; there was no measure of awareness of the consequences of SUD. What was measured here was the self- stigma. Additionally, the increased awareness in this context sounds more like insight into the consequences of SUD.

Introduction

The use of alcohol use disorder (AUD), marijuana use, and SUD interchangeably in this section may be problematic. For instance, Line 70- 71; “problematic marijuana use may culminate in a SUD”; why not a cannabis use disorder (CUD)?

Measures

Line 122-126

The statement …we replaced the term “mental illness” with “substance use disorder” throughout the measure…is worrying. Were the psychometric properties of the new instrument, created by replacing terminology, ascertained or was it an assumption that they were acceptable? Was there a definition of the term SUD that was provided to all the participants, or were they expected to respond using their own individual definitions and interpretation of what SUD meant- not a standardized one? Was it actually meant to be AUD, CUD or the actual DSM-5 TR SUD? Is there a possibility that the participants understood it to mean opioids, cocaine, nicotine, inhalants, hallucinogen and other substance use disorders? Was this how it was intended to be understood? Did SUD mean severity? Was a pilot on the instrument ever done after changing the terminology, to establish reliability and validity? So, do we actually know what was measured by this novel measure/ instrument and what the content of the results we are discussing here are?

If this was the thrust of the study, why not use ‘perception of self- stigma associated with developing a SUD in the context of substance use’, instead of alcohol or cannabis use? Alternatively, use ‘perception of self- stigma associated with developing an AUD/ CUD in the context of their use’. This would enable us to confidently compare “apples” with “apples” and not “oranges”.

Results

From the presentation of the results, it is clear that the participants’ socio-demographic characteristics were not considered, even with the obvious gender imbalance. Despite the fact that this was not part of the objectives, it would have been nice to know their residence, living arrangements, finances, marital status, educational level of study, age, and other characteristics that could be factors associated with the outcome.

Line 208- 209

Table 1- I suggest that the “significant differences” column is given a figure/ value/ number, to denote and support the significance level. The explanation given in the table is a repetition (see text in Line 203- 207) and should be removed.

Line 222- 223

Table 2- Were there scatter plots done to establish linearity? Does 1,2, 3…. columns represent the corresponding 1,2, 3…. rows in the bi-variate correlations? If so, it does not reflect the expected quality that a variable is always perfectly correlated with itself (r=1). May require clarification.

Discussion

This is largely determined by the findings; I have highlighted the concern in the findings. However, it was easy to notice that the limitations and future research section (Line 260- 309) was twice as long as the discussion (238- 258). Separating the limitations from the future research, so that they are two distinct sections could improve clarity. Future research begins from Line 270; although there is a small section, Line 277- 281 which is mainly focused on limitations.

7. PLOS authors have the option to publish the peer review history of their article (what does this mean?). If published, this will include your full peer review and any attached files.

Reviewer #2: No

Reviewer #3: No

Reviewer #5: No

Reviewer #8: No

---

## [Author Response · Author response to Decision Letter 2]

27 Feb 2024

Dr. Lakshit Jain,

We would like to thank you again for the thorough review of our manuscript and for the selection of well-qualified reviewers. We have addressed the reviewer’s concerns and summarized our changes below. By addressing the general journal requirements and reviewers’ comments, we believe our manuscript has been strengthened and will make more of a contribution to the field.

Editor Comments:

1. Three reviewers have recommend to accept this article, but reviewer 8 is seeking this article for the first time, they have raised multiple concerns that should be resolved prior to accepting this article

** We have addressed each reviewer 8’s concerns and summarized our changes below. By addressing the reviewer’s comments, we believe our manuscript has been strengthened and will make more of a contribution to the field.

Reviewer(s)' Response to Questions:

1. If the authors have adequately addressed your comments raised in a previous round of review and you feel that this manuscript is now acceptable for publication, you may indicate that here to bypass the “Comments to the Author” section, enter your conflict of interest statement in the “Confidential to Editor” section, and submit your "Accept" recommendation.

Reviewer #2: All comments have been addressed

Reviewer #3: All comments have been addressed

Reviewer #5: All comments have been addressed

Reviewer #8: (No Response)

** No action is required. 

2. Is the manuscript technically sound, and do the data support the conclusions? 

Reviewer #2: Yes

Reviewer #3: Yes

Reviewer #5: Yes

Reviewer #8: Partly

** No action is required. 

3. Has the statistical analysis been performed appropriately and rigorously?

Reviewer #2: Yes

Reviewer #3: Yes

Reviewer #5: Yes

Reviewer #8: I Don't Know

** No action is required. 

4. Have the authors made all data underlying the findings in their manuscript fully available?

Reviewer #2: Yes

Reviewer #3: Yes

Reviewer #5: Yes

Reviewer #8: Yes

** No action is required.

5. Is the manuscript presented in an intelligible fashion and written in standard English?

Reviewer #2: Yes

Reviewer #3: Yes

Reviewer #5: Yes

Reviewer #8: Yes

** No action is required. 

Review Comments to the Author:

Reviewer #2: (No Response)

** We appreciate the laudatory remarks. We thank the reviewer for their insightful comments, which we believe improved our article significantly.

Reviewer #3: The authors have sufficiently and succinctly made appropriate changes to the manuscript to make it more reader friendly. The paper presents an interesting approach to alcohol and cannabis use disorder propagation and at this stage would benefit from acceptance in the journal. Thank you for the opportunity to review this paper.

** We appreciate the laudatory remarks. We thank the reviewer for their insightful comments, which we believe improved our article significantly.

Reviewer #5: The paper has improved after the revisions. The authors' meticulous efforts have substantially strengthened the manuscript, aligning it more closely with the standards of PLOS ONE.

** We appreciate the laudatory remarks. We thank the reviewer for their insightful comments, which we believe improved our article significantly.

Reviewer #8: This manuscript presents information on college students’ perception of self- stigma associated with developing a substance use disorder (SUD) in the context of alcohol and marijuana use. It is well written and below are my suggestions on how the manuscript can be improved.

**We thank the reviewer for their insightful comments, which we believe improved our article significantly.

Abstract

Generally, the abstract would benefit from a structure; introduction of different sections will improve focus and clarity

**We have updated the abstract to make it more structured. Specifically, we have now added qualifiers for each section: background, aims, method, results, and conclusions. We hope these changes will improve the focus and clarity of our abstract section of the paper.

Line 32

There is no mention of the instruments used to collect the data.

**Given word limitations by the journal for abstracts, we did not want to write out every measure that was used to assess study constructs as it would push us past the word limit for the abstract based on the journal requirements.

Line 34 – 40

The results in this section require the support of the numbers that tell the story. Presentation could benefit by stating the actual figures obtained in the statistical analysis, including the p values while mentioning the significance levels.

For instance, were the higher levels of self- stigma mentioned in Line 37- 38 statistically significant? What were the actual figures?

**While we normally agree with this point, there are numerous correlations that would need to be presented for each outcome. Given the journal requirements regarding word limit for an abstract, including all of these statistics would push us past that limit. Thus, why they are omitted, and we defer to qualitative explanations of study results.

Line 44- 46

“Rather, we interpret these findings to indicate that increased awareness of the consequences of substance use disorder on the sense of self has an impact on alcohol and marijuana use among young adults”. When one has the objectives of the study in mind, it appears the different patterns of alcohol and marijuana use are actually the independent variables with self- stigma, and not awareness, as the dependent variable. This may require clarification.

The conclusion of “increased awareness” cannot emanate from the findings of this particular study; there was no measure of awareness of the consequences of SUD. What was measured here was the self- stigma. Additionally, the increased awareness in this context sounds more like insight into the consequences of SUD.

**We appreciate this comment and agree with the reviewer. We have rephrased this sentence such that it now reads, “Rather, we interpret these findings to indicate that negative perceptions of substance use disorder on the sense of self may be associated with distinct alcohol and marijuana use behaviors among young adults”.

Introduction

The use of alcohol use disorder (AUD), marijuana use, and SUD interchangeably in this section may be problematic. For instance, Line 70- 71; “problematic marijuana use may culminate in a SUD”; why not a cannabis use disorder (CUD)?

**We have carefully updated the introduction to be more focused on when we are describing AUD, CUD, and SUD.

Measures

Line 122-126

The statement …we replaced the term “mental illness” with “substance use disorder” throughout the measure…is worrying. Were the psychometric properties of the new instrument, created by replacing terminology, ascertained or was it an assumption that they were acceptable? Was there a definition of the term SUD that was provided to all the participants, or were they expected to respond using their own individual definitions and interpretation of what SUD meant- not a standardized one? Was it actually meant to be AUD, CUD or the actual DSM-5 TR SUD? Is there a possibility that the participants understood it to mean opioids, cocaine, nicotine, inhalants, hallucinogen and other substance use disorders? Was this how it was intended to be understood? Did SUD mean severity? Was a pilot on the instrument ever done after changing the terminology, to establish reliability and validity? So, do we actually know what was measured by this novel measure/ instrument and what the content of the results we are discussing here are?

**The reviewer raises several good points. Regarding the validity of this change to the prior measure, we wrote in the paragraph describing this measure that we conducted EFA analyses to examine its factor structure. We also provided more information to these analyses in S1 Appendix and we refer the reviewer to that section for more clarity of the reliability/validity of the measure.

Regarding the conceptual aspects, we purposely wanted to get people’s perceptions of SUD broadly and not tied to one drug. To that end, we purposely did not define SUD for participants as this definition varies across drug types and even cultures. Nonetheless, there is a possibility that this created issues in how people responded if they were focusing on one particular drug (e.g. opioids). We touch upon each of these issues in the limitations section and suggest ways to improve upon our study in future research (see pgs. 12-14). 

If this was the thrust of the study, why not use ‘perception of self- stigma associated with developing a SUD in the context of substance use’, instead of alcohol or cannabis use? Alternatively, use ‘perception of self- stigma associated with developing an AUD/ CUD in the context of their use’. This would enable us to confidently compare “apples” with “apples” and not “oranges”.

**Our original title was “Exploring Perceptions of Self-Stigma of Substance Use and Current Substance Use Patterns among College Students”. However, a different reviewer felt that it was too broad considering we only assessed alcohol and cannabis variables in correlating with SUD stigma perception. Thus, why the title is currently: “Exploring perceptions of self-stigma of substance use and current alcohol and marijuana use patterns among college students”.

Nonetheless, the reviewer is correct that future work should be tailored to specific perceptions of use disorder for a specific drug. Further, examining stigma perceptions of that particular drug (e.g., AUD) and how those perceptions associate with that drug use (e.g., alcohol use patterns) is important.

Results

From the presentation of the results, it is clear that the participants’ socio-demographic characteristics were not considered, even with the obvious gender imbalance. Despite the fact that this was not part of the objectives, it would have been nice to know their residence, living arrangements, finances, marital status, educational level of study, age, and other characteristics that could be factors associated with the outcome.

**The reviewer is correct in that our focus was not to examine sociodemographic predictors of perceptions of stigma related to SUD. We do describe some demographic information (age, gender, race/ethnicity) about the sample within the “participants and procedure” section. We don’t have information on marital status, residence, and finances which is why this information was not reported.

Line 208- 209

Table 1- I suggest that the “significant differences” column is given a figure/ value/ number, to denote and support the significance level. The explanation given in the table is a repetition (see text in Line 203- 207) and should be removed.

**Given that we already present the statistics about the omnibus ANOVA results in text, we did not feel it was appropriate in Table 1. Furthermore, we believe it is easier for readers who are not savvy in statistics to be able to digest and interpret the significant statistical results in a clearer manner. Further, we believe by reporting in the note section how statistical significance was determined, it can place in context our results for readers.

Line 222- 223

Table 2- Were there scatter plots done to establish linearity? Does 1,2, 3…. columns represent the corresponding 1,2, 3…. rows in the bi-variate correlations? If so, it does not reflect the expected quality that a variable is always perfectly correlated with itself (r=1). May require clarification.

**As mentioned in the note section of Table 2: Cronbach’s alpha for each measure is underlined and shown on the diagonal. We feel like this is more important information than just showing that each variable correlates perfectly with itself.

Discussion

This is largely determined by the findings; I have highlighted the concern in the findings. However, it was easy to notice that the limitations and future research section (Line 260- 309) was twice as long as the discussion (238- 258). Separating the limitations from the future research, so that they are two distinct sections could improve clarity. Future research begins from Line 270; although there is a small section, Line 277- 281 which is mainly focused on limitations

**We appreciate this point. However, we believe that by raising the limitations and then directly mentioning how these limitations can be improved upon in future research is easier for readers to digest. In other words, we did not want to list many limitations and then jump into a new section focused on future research that jumps around as opposed to focusing on how that future direction can alleviate current limitations of the present study.

---

## [Decision Letter · Decision Letter 3]

8 Mar 2024

PONE-D-23-05413R3Exploring perceptions of self-stigma of substance use and current alcohol and marijuana use patterns among college studentsPLOS ONE

Dear Dr. Bravo,

Thank you for submitting your manuscript to PLOS ONE. After careful consideration, we feel that it has merit but does not fully meet PLOS ONE’s publication criteria as it currently stands. Therefore, we invite you to submit a revised version of the manuscript that addresses the points raised during the review process.

\\**While 2 reviewers recommed accepting this Manuscript, Reviewer 8 has raised severela concerns. The reviewer also feels that many of the concerns they raised in their previous review have not been addressed. please see the review and address these concerns.** This manuscript presents information on college students’ perception of self- stigma associated with developing a substance use disorder (SUD) in the context of alcohol and marijuana use. It is well written and below are my suggestions on how the manuscript can be improved.

Abstract

Generally, the abstract would benefit from a structure; introduction of different sections will improve focus and clarity

Line 32

There is no mention of the instruments used to collect the data.

Line 34 – 40

The results in this section require the support of the numbers that tell the story. Presentation could benefit by stating the actual figures obtained in the statistical analysis, including the p values while mentioning the significance levels.

For instance, were the higher levels of self- stigma mentioned in Line 37- 38 statistically significant? What were the actual figures?

Line 44- 46

“Rather, we interpret these findings to indicate that increased awareness of the consequences of substance use disorder on the sense of self has an impact on alcohol and marijuana use among young adults”. When one has the objectives of the study in mind, it appears the different patterns of alcohol and marijuana use are actually the independent variables with self- stigma, and not awareness, as the dependent variable. This may require clarification.

The conclusion of “increased awareness” cannot emanate from the findings of this particular study; there was no measure of awareness of the consequences of SUD. What was measured here was the self- stigma. Additionally, the increased awareness in this context sounds more like insight into the consequences of SUD.

Introduction

The use of alcohol use disorder (AUD), marijuana use, and SUD interchangeably in this section may be problematic. For instance, Line 70- 71; “problematic marijuana use may culminate in a SUD”; why not a cannabis use disorder (CUD)?

Measures

Line 122-126

The statement …we replaced the term “mental illness” with “substance use disorder” throughout the measure…is worrying. Were the psychometric properties of the new instrument, created by replacing terminology, ascertained or was it an assumption that they were acceptable? Was there a definition of the term SUD that was provided to all the participants, or were they expected to respond using their own individual definitions and interpretation of what SUD meant- not a standardized one? Was it actually meant to be AUD, CUD or the actual DSM-5 TR SUD? Is there a possibility that the participants understood it to mean opioids, cocaine, nicotine, inhalants, hallucinogen and other substance use disorders? Was this how it was intended to be understood? Did SUD mean severity? Was a pilot on the instrument ever done after changing the terminology, to establish reliability and validity? So, do we actually know what was measured by this novel measure/ instrument and what the content of the results we are discussing here are?

If this was the thrust of the study, why not use ‘perception of self- stigma associated with developing a SUD in the context of substance use’, instead of alcohol or cannabis use? Alternatively, use ‘perception of self- stigma associated with developing an AUD/ CUD in the context of their use’. This would enable us to confidently compare “apples” with “apples” and not “oranges”.

Results

From the presentation of the results, it is clear that the participants’ socio-demographic characteristics were not considered, even with the obvious gender imbalance. Despite the fact that this was not part of the objectives, it would have been nice to know their residence, living arrangements, finances, marital status, educational level of study, age, and other characteristics that could be factors associated with the outcome.

Line 208- 209

Table 1- I suggest that the “significant differences” column is given a figure/ value/ number, to denote and support the significance level. The explanation given in the table is a repetition (see text in Line 203- 207) and should be removed.

Line 222- 223

Table 2- Were there scatter plots done to establish linearity? Does 1,2, 3…. columns represent the corresponding 1,2, 3…. rows in the bi-variate correlations? If so, it does not reflect the expected quality that a variable is always perfectly correlated with itself (r=1). May require clarification.

Discussion

This is largely determined by the findings; I have highlighted the concern in the findings. However, it was easy to notice that the limitations and future research section (Line 260- 309) was twice as long as the discussion (238- 258). Separating the limitations from the future research, so that they are two distinct sections could improve clarity. Future research begins from Line 270; although there is a small section, Line 277- 281 which is mainly focused on limitations.==============================

We look forward to receiving your revised manuscript.

Kind regards,

Lakshit Jain, MD

Academic Editor

PLOS ONE

Reviewers' comments:

Reviewer's Responses to Questions

**Comments to the Author**

1. If the authors have adequately addressed your comments raised in a previous round of review and you feel that this manuscript is now acceptable for publication, you may indicate that here to bypass the “Comments to the Author” section, enter your conflict of interest statement in the “Confidential to Editor” section, and submit your "Accept" recommendation.

Reviewer #3: All comments have been addressed

Reviewer #5: All comments have been addressed

Reviewer #8: (No Response)

2. Is the manuscript technically sound, and do the data support the conclusions?

Reviewer #3: Yes

Reviewer #5: Yes

Reviewer #8: Partly

3. Has the statistical analysis been performed appropriately and rigorously? 

Reviewer #3: Yes

Reviewer #5: Yes

Reviewer #8: I Don't Know

4. Have the authors made all data underlying the findings in their manuscript fully available?

Reviewer #3: Yes

Reviewer #5: Yes

Reviewer #8: Yes

5. Is the manuscript presented in an intelligible fashion and written in standard English?

Reviewer #3: Yes

Reviewer #5: Yes

Reviewer #8: Yes

6. Review Comments to the Author

Reviewer #3: The authors have sufficiently addressed and improved the manuscript in order for it to meet publication requirements and with the current revision I recommend that the paper be accepted for publications

Reviewer #5: The authors diligently addressed all comments provided by reviewers, ensuring clarity, coherence, and accuracy throughout the manuscript. Each comment was carefully considered, and revisions were made accordingly to enhance the overall quality of the work. This meticulous approach reflects the authors' commitment to producing a comprehensive and thoroughly vetted piece of scholarly literature.

Reviewer #8: The manuscript has really improved after the revisions. It is aligned closer to the standards of PLOS ONE.

7. PLOS authors have the option to publish the peer review history of their article (what does this mean?). If published, this will include your full peer review and any attached files.

Reviewer #3: No

Reviewer #5: **Yes: **Vinod Sharma

Reviewer #8: No

---

## [Author Response · Author response to Decision Letter 3]

13 Mar 2024

Dr. Lakshit Jain,

We would like to thank you again for the thorough review of our manuscript and for the selection of well-qualified reviewers. We have addressed the reviewer’s concerns and summarized our changes below. By addressing the general journal requirements and reviewers’ comments, we believe our manuscript has been strengthened and will make more of a contribution to the field.

Editor Comments:

1. Three reviewers have recommend to accept this article, but reviewer 8 is seeking this article for the first time, they have raised multiple concerns that should be resolved prior to accepting this article

** We have addressed each reviewer 8’s concerns and summarized our changes below. By addressing the reviewer’s comments, we believe our manuscript has been strengthened and will make more of a contribution to the field.

Reviewer(s)' Response to Questions:

1. If the authors have adequately addressed your comments raised in a previous round of review and you feel that this manuscript is now acceptable for publication, you may indicate that here to bypass the “Comments to the Author” section, enter your conflict of interest statement in the “Confidential to Editor” section, and submit your "Accept" recommendation.

Reviewer #2: All comments have been addressed

Reviewer #3: All comments have been addressed

Reviewer #5: All comments have been addressed

Reviewer #8: (No Response)

** No action is required. 

2. Is the manuscript technically sound, and do the data support the conclusions? 

Reviewer #2: Yes

Reviewer #3: Yes

Reviewer #5: Yes

Reviewer #8: Partly

** No action is required. 

3. Has the statistical analysis been performed appropriately and rigorously?

Reviewer #2: Yes

Reviewer #3: Yes

Reviewer #5: Yes

Reviewer #8: I Don't Know

** No action is required. 

4. Have the authors made all data underlying the findings in their manuscript fully available?

Reviewer #2: Yes

Reviewer #3: Yes

Reviewer #5: Yes

Reviewer #8: Yes

** No action is required.

5. Is the manuscript presented in an intelligible fashion and written in standard English?

Reviewer #2: Yes

Reviewer #3: Yes

Reviewer #5: Yes

Reviewer #8: Yes

** No action is required. 

Review Comments to the Author:

Reviewer #2: (No Response)

** We appreciate the laudatory remarks. We thank the reviewer for their insightful comments, which we believe improved our article significantly.

Reviewer #3: The authors have sufficiently and succinctly made appropriate changes to the manuscript to make it more reader friendly. The paper presents an interesting approach to alcohol and cannabis use disorder propagation and at this stage would benefit from acceptance in the journal. Thank you for the opportunity to review this paper.

** We appreciate the laudatory remarks. We thank the reviewer for their insightful comments, which we believe improved our article significantly.

Reviewer #5: The paper has improved after the revisions. The authors' meticulous efforts have substantially strengthened the manuscript, aligning it more closely with the standards of PLOS ONE.

** We appreciate the laudatory remarks. We thank the reviewer for their insightful comments, which we believe improved our article significantly.

Reviewer #8: This manuscript presents information on college students’ perception of self- stigma associated with developing a substance use disorder (SUD) in the context of alcohol and marijuana use. It is well written and below are my suggestions on how the manuscript can be improved.

**We thank the reviewer for their insightful comments, which we believe improved our article significantly.

Abstract

Generally, the abstract would benefit from a structure; introduction of different sections will improve focus and clarity

**We have updated the abstract to make it more structured. Specifically, we have now added qualifiers for each section: background, aims, method, results, and conclusions. We hope these changes will improve the focus and clarity of our abstract section of the paper.

Line 32

There is no mention of the instruments used to collect the data.

**Given word limitations by the journal for abstracts, we did not want to write out every measure that was used to assess study constructs as it would push us past the word limit for the abstract based on the journal requirements.

Line 34 – 40

The results in this section require the support of the numbers that tell the story. Presentation could benefit by stating the actual figures obtained in the statistical analysis, including the p values while mentioning the significance levels.

For instance, were the higher levels of self- stigma mentioned in Line 37- 38 statistically significant? What were the actual figures?

**While we normally agree with this point, there are numerous correlations that would need to be presented for each outcome. Given the journal requirements regarding word limit for an abstract, including all of these statistics would push us past that limit. Thus, why they are omitted, and we defer to qualitative explanations of study results.

Line 44- 46

“Rather, we interpret these findings to indicate that increased awareness of the consequences of substance use disorder on the sense of self has an impact on alcohol and marijuana use among young adults”. When one has the objectives of the study in mind, it appears the different patterns of alcohol and marijuana use are actually the independent variables with self- stigma, and not awareness, as the dependent variable. This may require clarification.

The conclusion of “increased awareness” cannot emanate from the findings of this particular study; there was no measure of awareness of the consequences of SUD. What was measured here was the self- stigma. Additionally, the increased awareness in this context sounds more like insight into the consequences of SUD.

**We appreciate this comment and agree with the reviewer. We have rephrased this sentence such that it now reads, “Rather, we interpret these findings to indicate that negative perceptions of substance use disorder on the sense of self may be associated with distinct alcohol and marijuana use behaviors among young adults”.

Introduction

The use of alcohol use disorder (AUD), marijuana use, and SUD interchangeably in this section may be problematic. For instance, Line 70- 71; “problematic marijuana use may culminate in a SUD”; why not a cannabis use disorder (CUD)?

**We have carefully updated the introduction to be more focused on when we are describing AUD, CUD, and SUD.

Measures

Line 122-126

The statement …we replaced the term “mental illness” with “substance use disorder” throughout the measure…is worrying. Were the psychometric properties of the new instrument, created by replacing terminology, ascertained or was it an assumption that they were acceptable? Was there a definition of the term SUD that was provided to all the participants, or were they expected to respond using their own individual definitions and interpretation of what SUD meant- not a standardized one? Was it actually meant to be AUD, CUD or the actual DSM-5 TR SUD? Is there a possibility that the participants understood it to mean opioids, cocaine, nicotine, inhalants, hallucinogen and other substance use disorders? Was this how it was intended to be understood? Did SUD mean severity? Was a pilot on the instrument ever done after changing the terminology, to establish reliability and validity? So, do we actually know what was measured by this novel measure/ instrument and what the content of the results we are discussing here are?

**The reviewer raises several good points. Regarding the validity of this change to the prior measure, we wrote in the paragraph describing this measure that we conducted EFA analyses to examine its factor structure. We also provided more information to these analyses in S1 Appendix and we refer the reviewer to that section for more clarity of the reliability/validity of the measure.

Regarding the conceptual aspects, we purposely wanted to get people’s perceptions of SUD broadly and not tied to one drug. To that end, we purposely did not define SUD for participants as this definition varies across drug types and even cultures. Nonetheless, there is a possibility that this created issues in how people responded if they were focusing on one particular drug (e.g. opioids). We touch upon each of these issues in the limitations section and suggest ways to improve upon our study in future research (see pgs. 12-14). 

If this was the thrust of the study, why not use ‘perception of self- stigma associated with developing a SUD in the context of substance use’, instead of alcohol or cannabis use? Alternatively, use ‘perception of self- stigma associated with developing an AUD/ CUD in the context of their use’. This would enable us to confidently compare “apples” with “apples” and not “oranges”.

**Our original title was “Exploring Perceptions of Self-Stigma of Substance Use and Current Substance Use Patterns among College Students”. However, a different reviewer felt that it was too broad considering we only assessed alcohol and cannabis variables in correlating with SUD stigma perception. Thus, why the title is currently: “Exploring perceptions of self-stigma of substance use and current alcohol and marijuana use patterns among college students”.

Nonetheless, the reviewer is correct that future work should be tailored to specific perceptions of use disorder for a specific drug. Further, examining stigma perceptions of that particular drug (e.g., AUD) and how those perceptions associate with that drug use (e.g., alcohol use patterns) is important.

Results

From the presentation of the results, it is clear that the participants’ socio-demographic characteristics were not considered, even with the obvious gender imbalance. Despite the fact that this was not part of the objectives, it would have been nice to know their residence, living arrangements, finances, marital status, educational level of study, age, and other characteristics that could be factors associated with the outcome.

**The reviewer is correct in that our focus was not to examine sociodemographic predictors of perceptions of stigma related to SUD. We do describe some demographic information (age, gender, race/ethnicity) about the sample within the “participants and procedure” section. We don’t have information on marital status, residence, and finances which is why this information was not reported.

Line 208- 209

Table 1- I suggest that the “significant differences” column is given a figure/ value/ number, to denote and support the significance level. The explanation given in the table is a repetition (see text in Line 203- 207) and should be removed.

**Given that we already present the statistics about the omnibus ANOVA results in text, we did not feel it was appropriate in Table 1. Furthermore, we believe it is easier for readers who are not savvy in statistics to be able to digest and interpret the significant statistical results in a clearer manner. Further, we believe by reporting in the note section how statistical significance was determined, it can place in context our results for readers.

Line 222- 223

Table 2- Were there scatter plots done to establish linearity? Does 1,2, 3…. columns represent the corresponding 1,2, 3…. rows in the bi-variate correlations? If so, it does not reflect the expected quality that a variable is always perfectly correlated with itself (r=1). May require clarification.

**The reviewer is correct in that constructs correlate perfectly (i.e., r = 1) with itself. Given such common knowledge, and as mentioned in the note section of Table 2, we instead report on Cronbach’s alpha for each measure and these values are underlined and shown on the diagonal in the Table. We feel like this is more important information than just showing that each variable correlates perfectly with itself.

Discussion

This is largely determined by the findings; I have highlighted the concern in the findings. However, it was easy to notice that the limitations and future research section (Line 260- 309) was twice as long as the discussion (238- 258). Separating the limitations from the future research, so that they are two distinct sections could improve clarity. Future research begins from Line 270; although there is a small section, Line 277- 281 which is mainly focused on limitations

**We appreciate this point. However, we believe that by raising the limitations and then directly mentioning how these limitations can be improved upon in future research is easier for readers to digest. In other words, we did not want to list many limitations and then jump into a new section focused on future research that jumps around as opposed to focusing on how that future direction can alleviate current limitations of the present study.

---

## [Editor Report · Decision Letter 4]

18 Mar 2024

Exploring perceptions of self-stigma of substance use and current alcohol and marijuana use patterns among college students

PONE-D-23-05413R4

Dear Dr. Bravo,

We’re pleased to inform you that your manuscript has been judged scientifically suitable for publication and will be formally accepted for publication once it meets all outstanding technical requirements.

Kind regards,

Lakshit Jain, MD

Academic Editor

PLOS ONE
---

## [Editor Report · Acceptance letter]

27 Mar 2024

PONE-D-23-05413R4 

PLOS ONE

Dear Dr. Bravo, 

I'm pleased to inform you that your manuscript has been deemed suitable for publication in PLOS ONE. Congratulations! Your manuscript is now being handed over to our production team.

Kind regards, 

on behalf of

Dr. Lakshit Jain 

Academic Editor

PLOS ONE